# The Role of the Pancreatic Stone Protein in Predicting Intra-Abdominal Infection-Related Complications: A Prospective Observational Single-Center Cohort Study

**DOI:** 10.3390/microorganisms11102579

**Published:** 2023-10-17

**Authors:** Christos Michailides, Maria Lagadinou, Themistoklis Paraskevas, Konstantinos Papantoniou, Michael Kavvousanos, Achilleas Vasileiou, Konstantinos Thomopoulos, Dimitrios Velissaris, Markos Marangos

**Affiliations:** 1Department of Internal Medicine, University Hospital of Patras, 26504 Patras, Greece; mlagad@upatras.gr (M.L.); themispara@hotmail.com (T.P.); guspapanton@yahoo.gr (K.P.); mixalis.kav.1995@gmail.com (M.K.); dvelissaris@upatras.gr (D.V.); marangos@upatras.gr (M.M.); 2Department of Gastroenterology, University Hospital of Patras, 26504 Patras, Greece; kxthomo@hotmail.com

**Keywords:** biomarkers, intrabdominal infections, prognosis

## Abstract

Background: The Pancreatic Stone Protein (PSP) is an acute-phase protein that is mainly secreted by pancreatic cells in response to stress. The current literature supports its use as a predictor of sepsis. Its prognostic role has recently been evaluated in a point-of-care setting, mostly in high-risk patients. We conducted a prospective observational cohort study to evaluate its utility in the prognosis of patients admitted to the hospital with a diagnosis of intra-abdominal infection. Methods: Adult patients consecutively admitted to the Internal Medicine Department of the University Hospital of Patras, Greece, with a diagnosis of intra-abdominal infection were enrolled. PSP levels were measured within 24 h of admission in whole blood. Results: a total of 40 patients were included after being diagnosed with IAI. PSP was used as an independent predictive factor for sepsis after adjusting for age with OR = 7.888 (95% CI: 1.247–49.890). PSP also predicted readmission and the need for treatment escalation (*p*: <0.01) and was an excellent prognostic factor regarding these outcomes (AUC = 0.899, 95% CI: 0.794–1.0, and AUC = 0.862, 95% CI: 0.748–0.976, respectively). PSP also proved superior to CRP, ferritin, and fibrinogen in sepsis diagnosis, treatment escalation, and readmission prediction with an AUC of 0.862, 0.698, and 0.899, respectively. Conclusions: PSP can predict unfavorable outcomes, such as sepsis development, readmission, and the need for treatment escalation among patients with intra-abdominal infections.

## 1. Introduction

Intra-abdominal infections (IAIs) define a heterogenous group of diseases with a wide range of phenotypes, from mild disease to septic conditions with significant morbidity and mortality. Source control and targeted antimicrobial therapy are ground principles to treat such infections [1]. Life-threatening conditions that may need specific interventions, such as surgery, escalated antibiotic treatment, or management in the intensive care setting, must be distinguished as early as possible. Several biomarkers have been evaluated for their utility in the early diagnosis and prediction of clinical outcomes in patients with infection and sepsis. In 2004, the Pancreatic Stone Protein (PSP) was first introduced as a diagnostic and prognostic biomarker for sepsis and SIRS [2]. PSP is a protein mainly produced in the exocrine pancreas and primarily described in the pathogenesis of pancreatitis. Its secretion is also triggered by systematic inflammation and is then produced in several organs, leading to a peak in the acute phase of inflammation [3]. Furthermore, PSP is a pro-inflammatory molecule that binds to polymorphonuclear (PMN) cells and triggers their activation [4]. It is still unknown whether PSP is superior to other established and easily measured biomarkers, such as WBC and CRP, in predicting the outcome of patients who present with an abdominal infection.

The aim of this study is to assess the prognostic accuracy of PSP in predicting intra-abdominal infection-related complications and disease course, namely the need for antibiotic escalation, ICU admission, hospital readmission, surgery, and outcomes. Additionally, we aim to elucidate its superiority compared to previously used biomarkers in a point-of-care setting.

## 2. Materials and Methods

### 2.1. Study Design

Adult patients (>18 years old) who were consecutively admitted to the Internal Medicine Department of the University Hospital of Patras, Greece, from March 2023 to June 2023, with signs and symptoms of any intra-abdominal infection, were enrolled. Written informed consent was obtained from each patient or the patient’s legally authorized representative. A trial was conducted in accordance with the International Conference on Harmonization E6 guidelines for Good Clinical Practice, the Declaration of Helsinki, and local regulations. The study was approved by the institutional review board (125/4 April 2023) and the local ethics committee of the University Hospital of Patras, Greece (94/16 March 2023).

### 2.2. Participants

Patients aged >18 years old who presented with signs and symptoms of any intra-abdominal infection (cholecysitis, cholangitis, diverticulitis, typhlitis, appendicitis, and contaminated necrotizing pancreatitis) were included in the analysis. We excluded patients with complicated co-infections on admission (endocarditis, Central Nervous System infection, septic arthritis, osteomyelitis, polymicrobial infections), those who were hospitalized or had received antibiotic treatment for the last month, including bacteremic patients with XDR or PDR microorganisms, and severely immunocompromised patients (neutropenic, patients diagnosed with hematological diseases or those with hematological or solid organ malignancies). All participants were tracked for 28 days to document their outcomes.

### 2.3. Outcome Measures

The primary endpoints were the length of hospital stay, the duration of antibiotic treatment, and sepsis or septic shock development. Secondary outcomes were patients’ final outcome (discharge or all-cause in-hospital mortality, mortality on days 14 and 28), need for treatment escalation, symptoms duration, and ICU admission.

### 2.4. Data Collection

All participants admitted to the ED between March 2023 and June 2023 and who met the inclusion criteria were subjected to single blood sampling right after the diagnosis of IAI and the decision for hospitalization. We recorded the basic laboratory tests as follows: full blood count, liver panel and electrolyte panel, lactate dehydrogonase (LDH), and acute phase proteins (ferritin, fibrinogen, CRP) to all participants. Additionally, we performed blood cultures and imaging exams (abdominal ultrasound (US) and abdominal computed tomography (CT)) on the admission day for each patient and, according to their clinical presentation, when needed to diagnose each infection. All patients received empirical antibiotic therapy with broad-spectrum antibiotics from day 1, followed by targeted antibiotics after the isolation of a pathogen.

After the diagnosis of intrabdominal infection was established, we measured Pancreatic Stone Protein (PSP) plasma levels on admission day from the single first 24 h of sampling. PSP was measured soon after sample collection in whole blood, using a diagnostic capsule for quantitative measurement and a desktop spectrophotometer diagnostic device (abioSCOPE). After a six-minute process, the result was delivered in ng/dL.

### 2.5. Statistical Methods

All statistical analysis was performed using IBM^®^ SPSS^®^ Statistic v.26 software. Normally distributed continuous data are presented as the mean, and nominal data are presented as absolute values and percentages. The Shapiro–Wilk test of normality was used. ROC curve analysis was performed to determine the prognostic accuracy of PSP for main and secondary outcomes [5]. Sensitivity and specificity were calculated at the optimal cut-off (based on the Youden Index) for each outcome, as well as for the optimal cut-off for the main outcome. No pre-defined threshold was used as there was not enough evidence based on the available literature. Spearman’s rank correlation test was used to explore the possible correlation between continuous variables. Mann–Whitney’s U test was used for hypothesis testing in non-normally distributed continuous variables.

To ensure proper patient selection, consecutive sample methods were chosen, and the case–control design was avoided. Additionally, the data analyst was blinded to the arbitrary assignment (0/1) of positive and negative outcomes.

### 2.6. Definitions

Intrabdominal infection is considered an infection of any organ or tissue cited below the diaphragm inside the abdominal cavity. Sepsis development is considered an increase in the SOFA score by 2 points or more. Septic shock is considered hemodynamic instability, which is caused by an infection that needs vasopressors to maintain normal blood pressure despite the proper fluid infusion or an increase in lactate acid over the value of 2 mmol/lt. The diagnosis of cholecystitis and cholangitis is was set by clinical and laboratory findings in combination with ultrasound, and the diagnosis of the remaining infections is was set using clinical signs and symptoms, laboratory findings, and a CT if needed.

## 3. Results

A total of 40 patients with intra-abdominal infections were included in this study. The mean age of our cohort was 64.2 ± 22.8 years. Nineteen patients were male (47.5%). Co-morbidities were present in 37 patients (92.5%). Hypertension was observed as the most common co-morbidity (47.5%), followed by cardiovascular diseases (20%), hyperlipidemia (20%), hypothyroidism (17.5%), diabetes (10%), and malignancies (10%) Twenty-two patients (55%) were diagnosed with sepsis during their hospitalization, according to the sepsis-3 consensus definition. The median duration of hospitalization was 5 days. The most common type of infection was cholecystitis, followed by cholangitis. None of the patients included in our cohort died during hospitalization, and one patient died within the 30 d follow-up period. All the aforementioned information is shown in Table 1. Moreover, all primary and secondary outcomes of the study in relation to PSP measurements are summarized in Table 2. 

### 3.1. Main Outcomes

#### 3.1.1. Sepsis

In total, 22 out of 40 patients fulfilled the diagnostic criteria for sepsis according to sepsis-3 definitions during hospitalization. The median PSP on admission for patients with sepsis was 162 ng/dL (86.75–254.25) and 74.5 ng/dL (47.25–141.25) in patients without sepsis. There was a significant difference between the two groups (*p* = 0.037), but PSP only had a moderate prognostic accuracy for this outcome, with an AUC of 0.694 (95% CI: 0.525–0.861). The optimal cut-off was 71 ng/dL with a sensitivity of 90.9% and a specificity of 50% (Figure 1a,b).

Then, we calculated the Unadjusted Odds Ratio (OR) of PSP to predict sepsis, and it was 10.0 (95% CI: 1.786–55.976). Due to our aged population, we adjusted PSP with age, though it remained an independent prognostic factor for that outcome (*p*: 0.028, OR: 7.888) (95% CI: 1.2470–49.890) (Table 3).

#### 3.1.2. Length of Stay

PSP was not significantly correlated with the length of stay in our cohort (*p* = 0.103, r = −0.261) (Figure 2).

### 3.2. Secondary Outcomes

#### 3.2.1. Readmission

Data regarding readmission were available for 36 out of 40 patients, with 14 of them being readmitted to the hospital within 30 days of their first hospitalization. PSP on admission (of their first hospitalization) was significantly higher (*p*: <0.01) in patients who were readmitted compared to those who were not, with a median value of 203 ng/dL (158.75–376) and 71 ng/dL (46.5–107.75), respectively. PSP on admission had an excellent to outstanding prognostic value for this outcome (AUC = 0.899, 95% CI: 0.794–1.0). Sensitivity and specificity for the optimal cut-off point (120 ng/dL) were 85.7% and 86.4%, respectively (Figure 3).

#### 3.2.2. Treatment Escalation

Treatment escalation was required in 29 out of 40 patients either due to the persistence of fever beyond 48 h or due to the persistence or even worsening of symptoms. However, only one patient was treated surgically. PSP differed significantly between these two groups [86 (52–119.5) vs. 212 (175–388), *p* < 0.01]. PSP had an excellent prognostic value for this outcome (AUC = 0.862, 95% CI: 0.748–0.976). Sensitivity and specificity on the optimal cut-off point (162 ng/dL) were 90.9% and 73.7%, respectively (Figure 4).

#### 3.2.3. Invasive Treatment

We did not observe a difference between the patients who required an invasive treatment [median = 133 ng/dL, 54.75–221.25] and those who did not [median = 105 ng/dL, 67.25–212.75] (Figure 5).

#### 3.2.4. Comparison to Other Inflammatory Biomarkers

We did not observe a significant correlation between PSP and any other of the commonly used established biomarkers of inflammation (Ferritin, CRP, LDH, Fibrinogen). PSP was superior to the other biomarkers of inflammation for predicting sepsis, treatment escalation, and readmission (Table 4).

## 4. Discussion

The results of this study demonstrate that a measurement of PSP within the first 24 h since admission can predict unfavorable outcomes, such as the development of sepsis during hospitalization, the need to escalate antibiotic treatment, and the readmission of a patient within one month. To our knowledge, this is the first study to evaluate PSP as a prognostic factor for complications induced by intrabdominal infections (IAI). According to the published literature, the PSP value increases sharply and immediately after sepsis initiation [6,7]. The high sensitivity of PSP (>85%) to detect these outcomes could be a promising tool for clinicians to decide whether a patient needs hospitalization and intensive care treatment. After the diagnosis of IAI, a PSP measurement could guide the decision to dismiss or hospitalize the patient and could set an alarm for the possibility of close monitoring and advanced care. PSP’s superiority compared to other frequently used biomarkers in predicting such outcomes could also support its use in settings such as Emergency Departments (ED).

The pancreatic stone protein has been previously studied as a point-of-care biomarker used in an ED setting. Guadiana-Romualdo et al. measured PSP on admission to the ED in patients with possible infection. Patients were successfully categorized as either non-infected, infected, or septic [8]. A previous study by our team has set a cut-off value of 51 ng/mL on ED admission to predict prolonged hospitalization among COVID-19 patients [9]. A systematic review and meta-analysis by Prazak et al. demonstrated PSP as an optimal infection diagnostic tool [10].

Regarding its prognostic role in sepsis among those patients, its moderate but salutary accuracy, with an AUC of 0.694 (95% CI: 0.525–0.861), is in line with other studies that have examined this outcome in different settings. De Hond et al. found a similar AUC of 0.65 when predicting sepsis after excluding COVID-19 patients. In this study, PSP was also superior to WBCs and CRP in predicting sepsis [11]. At the cut-off point of 30 ng/mL, PSP has been described to discriminate septic patients from those with non-infective SIRS, with a sensitivity of 88% and a specificity of 78% [12]. In a setting of post-cardiac surgery, the evaluation of PSP on day 2 after surgery was found to predict postoperative infections with a sensitivity of 64% and a specificity of 70% at a cut-off value of 48.1 ng/mL, in contrast with CRP and WBCs that could not predict this outcome. Its sharper increase was evidence of sepsis development [13]. The mean value of PSP on admission in patients who had bacteremia was 133 ng/mL vs. 59 ng/mL in patients without bacteremia in a study that evaluated patients admitted to the ED [8]. These findings are similar to ours regarding the mean values of septic patients (PSP = 162 ng/mL) vs. non-septic patients (PSP = 74.5 ng/mL). Some studies have described the possible utility of PSP in predicting mortality. A day 2, a measurement was found to have a statistically significant difference between survivals and non-survivals, while a measurement within the first 24 h since ICU admission was demonstrated as a moderate predictor for mortality among critically ill and infected patients [14,15]. However, these data pertain to ICU patients who have an increased risk of mortality in contrast with patients who arrive at the ED with a later diagnosis of IAI and in whom the rates of mortality are almost 0%. PSP can further stratify septic patients in ICUs into those who develop MODS and those who are more likely to die [16].

PSP has also been studied as a prognostic factor for specific infections such as ventilator-associated pneumonia (VAP) and peritonitis. Among these patients admitted to the ICU, PSP can predict death and organ failure in a point-of-care setting [17,18]. Additionally, PSP seems to distinguish women with normal pregnancy from those with pregnancy complications [19]. Its implementation as a biomarker for IAIs is in line with our results, which inferentially append a new perspective to its usage.

There are several studies that compare PSP to other frequently used biomarkers, such as CRP and WBC. PSP seems to outclass them as a prognostic factor since it is the only marker that is proven to have the ability to predict postoperative infections among patients who undergo cardiac surgery, septic events among patients admitted to the ED, and patients with severe burn trauma [7,11,13]. PSP for point-of-care use has also been described to differentiate survivors and non-survivors among ICU COVID-19 patients in reiterative measurements, in addition to CRP and PCT [20]. We have established this superiority by comparing PSP with CRP, ferritin, fibrinogen, and WBC in terms of predicting unfavorable outcomes in patients with IAI. PSP has the best AUC for the prediction of sepsis, readmission, and treatment escalation (0.698, 0.899, and 0.872, respectively).

This study strengthens the current literature with new evidence of PSP application in clinical practice. A protein that is originally produced by the pancreas to avert stone formation and increases in inflammatory conditions and systemic stress due to its secretion from other organs such as the kidney, intestines, and stomach, can dramatize a significant role in the prognosis of IAI [21,22,23]. Unfavorable outcomes that increase public health costs, such as readmission and treatment escalation, are evaluated herein for the first time with promising results. The confirmation of previous data that PSP is a potent prognostic biomarker for sepsis, which is superior to other biomarkers, especially ferritin, and fibrinogen, that have never been compared before, demonstrates its potential for wider use. The fact that PSP remains an independent prognostic factor after adjusting for age amplifies this inference.

The limitations of our study include the determinate sample size, which did not allow us to collect sufficient data for mortality prognosis, and the inability of our surgery department to operate all the necessary invasive procedures due to extreme workload, which led us to treat most of the patients conservatively, with a direct impact on our results regarding invasive treatment and the incapability to measure PSP multiple times throughout our patients’ course. The absence of patients who died during hospitalization did not allow us to provide data for this outcome.

In conclusion, PSP is very promising for predicting unfavorable outcomes, especially in IAI, where its production normally originates. Distant infections may not release such a response despite PSP being considered an acute phase protein [23]. Its sharp increase after infection or sepsis onset classifies it as a point-of-care marker [11]. The usefulness of this study is that PSP plasma levels on admission could, additionally to the rest of clinical, laboratory, and imaging findings, help clinicians to classify patients according to PSP and decide whether a patient needs hospitalization or not and distinguish those who need closer monitoring for complications throughout their hospitalization. Thus, further investigation with a larger sample size and the participation of more centers is warranted to establish such an implementation. More studies with different populations are needed to navigate future meta-analyses to set worldwide cut-offs for infection and sepsis risk.

## Figures and Tables

**Figure 1 microorganisms-11-02579-f001:**
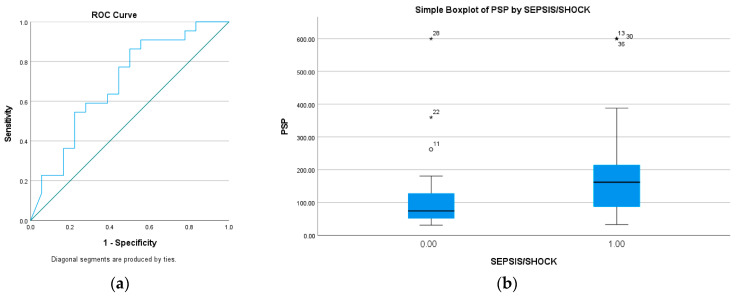
(**a**) ROC analysis of Pancreatic Stone Protein on prognostic accuracy of sepsis, (**b**) Measurements for PSP in patients with and without sepsis (162 ng/dL vs. 74.5 ng/dL).

**Figure 2 microorganisms-11-02579-f002:**
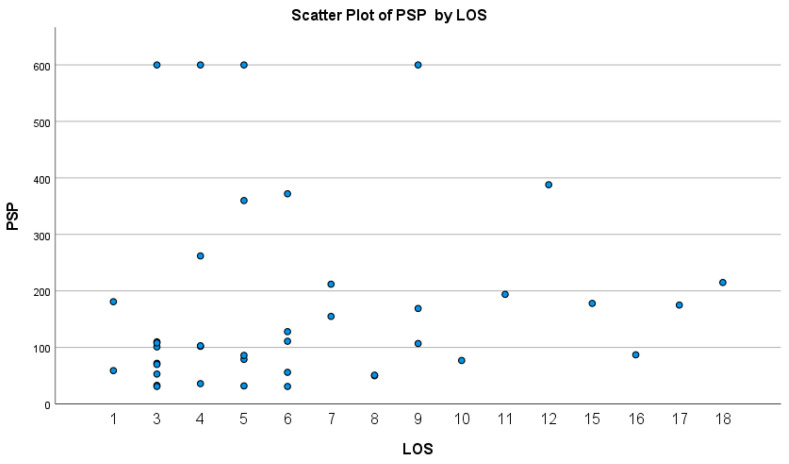
Scatter plot of PSP with length of hospitalization/stay (LOS).

**Figure 3 microorganisms-11-02579-f003:**
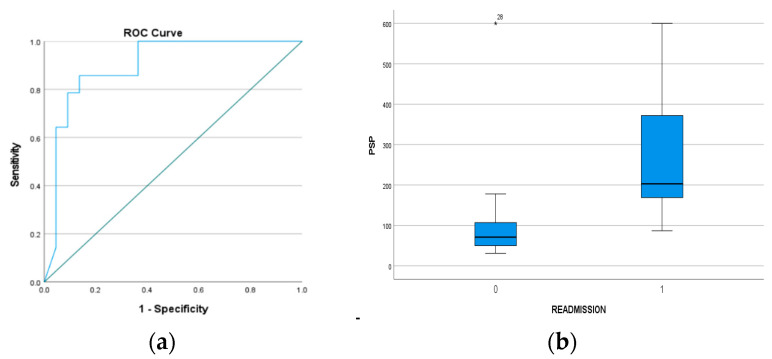
(**a**) ROC analysis of PSP on prognostic accuracy of readmission. (**b**) PSP on first admission was significantly higher in patients who were readmitted compared to those who were not, with a median value of 203 ng/dL vs. 71 ng/dL.

**Figure 4 microorganisms-11-02579-f004:**
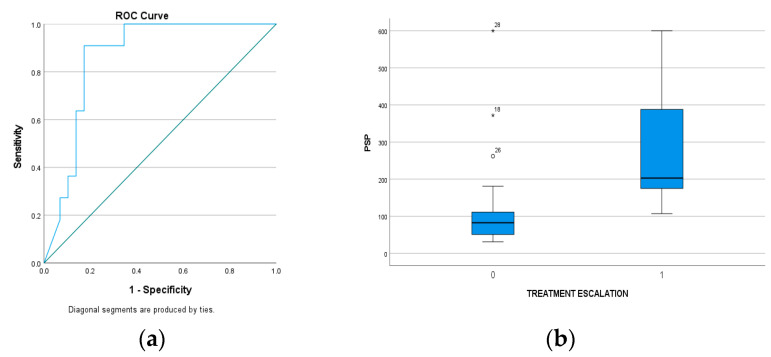
(**a**) ROC analysis of PSP with an excellent prognostic value for treatment escalation (AUC: 0.862), (**b**) PSP measurements differed statistically and significantly between the two groups of patients (treatment escalation and not): 212 ng/dL vs. 86 ng/dL.

**Figure 5 microorganisms-11-02579-f005:**
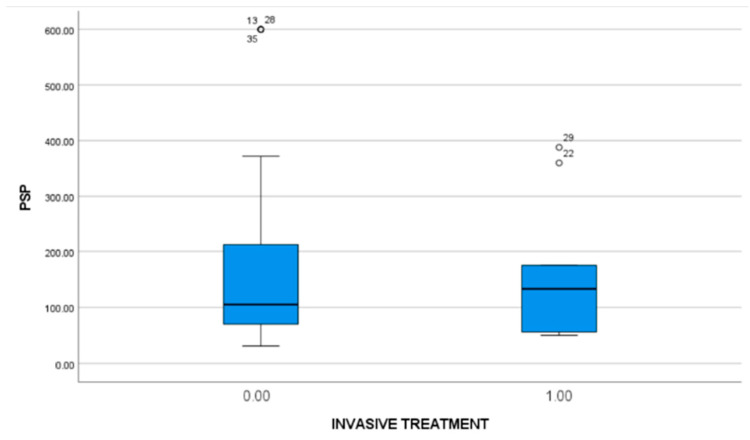
Invasive treatment: No statistically significant difference was found between the patients that required an invasive treatment (133 ng/dL) and those that did not (105 ng/dL).

**Table 1 microorganisms-11-02579-t001:** Epidemiological data, type of intrabdominal infection, and laboratory findings of the enrolled patients.

Age (Years)		64.2 ± 22.8
Comorbidities		
	Hypertension	19/40 (47.5%)
	Cardiovascular disease	8/40 (20%)
	Hypothyrodism	4/40 (10%)
	Hyperlipidemia	7/40 (17.5%)
	Diabetes	8/40 (20%)
	Malignancies	4/40 (10%)
		4/40 (10%)
**Site of Infection**		
	Gastroenteritis	5/40 (7.5%)
	Diverticulitis	6/40 (15%)
	Colitis	6/40 (15%)
	Mesenteric lymphadenitis	1/40 (2.5%)
	Necrotising pancreatitis	1/40 (2.5%)
	Appendicitis	1/40 (2.5%)
	Cholangitis	7/40 (17.5%)
	Cholecystitis	13/40 (32.5%)
**PSP (ng/dL)**		119 (61.75–250.25)
**Ferritin (mg/dL)**		214 (129.75–588.5)
**CRP (mg/dL)**		5.99 (0.73–16.67)
**Fibrinogen (mg/dL)**		489.75 (129.75–588.5)
**Mortality (In-hospital)**		0/40 (0%)
**Mortality (14 d)**		1/40 (2.5%)
**Mortality (28 d)**		1/40 (2.5%)
**Sepsis development**		22/40 (55%)
**Treatment Escalation**		11/40 (27.5%)
**Invasive Treatment**		10/40 (25%)
**Readmission**		14/36 (38.9%)
**Length of Stay (days)**		5 (3–8.75)

**Table 2 microorganisms-11-02579-t002:** Outcome measures in relation to PSP. The table below shows PSP differences and prognostic accuracy when predicting sepsis development, readmission, treatment escalation and invasive treatment.

Outcome	Mann–Whitney *p* Value	AUC	Cut-Off	Sensitivity	Specificity
Sepsis	0.037	0.693 (0.525–0.891)	71 (Optimal)	90.9%	50%
Readmission	0.050	0.734 (0.564–0.905)	71	100%	50%
			120 (Optimal)	85.7%	86.4%
Treatment Escalation	<0.001	0.862 (0.748–0.976)	71	100%	37.9%
			162 (Optimal)	90.9%	73.7%
Invasive Treatment	0.914	-	-	-	-

**Table 3 microorganisms-11-02579-t003:** Unadjusted Odds Ratio (OR) of PSP to predict sepsis.

							95% C.I for EXP(B)	
	B	S.E	Wald	df	Sig.	Exp(B)	Lower	Upper
**PSP**	2.065	0.941	4.816	1	0.028	7.888	1.247	49.890
**AGE**	0.042	0.020	4.677	1	0.031	1.043	1.004	1.084
**Constant**	−4.043	1.532	6.966	1	0.008	0.018		
**PSP**	2.303	0.879	6.866	1	0.009	10.000	1.786	55.976

**Table 4 microorganisms-11-02579-t004:** Comparison of PSP with other inflammatory biomarkers.

	Treatment Escalation		Sepsis		Readmission	
Inflammatory Biomarkers	Area	Std. Error	Area	Std. Error	Area	Std. Error
PSP (ng/dL)	0.862	0.058	0.698	0.86	0.899	0.053
Ferritin (mg/dL)	0.678	0.109	0.541	0.112	0.718	0.107
CRP (mg/dL)	0.575	0.098	0.533	0.093	0.628	0.098
Fibrinogen (mg/dL)	0.566	0.090	0.630	0.090	0.670	0.092

## Data Availability

The datasets generated and analyzed during the current study are available upon reasonable request to the corresponding author.

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
