# Peer review of "The Role of the Pancreatic Stone Protein in Predicting Intra-Abdominal Infection-Related Complications: A Prospective Observational Single-Center Cohort Study"

_microorganisms, 2023, doi:10.3390/microorganisms11102579_

Round 1
Reviewer 1 Report
Comments to Authors
This study showed that PSP can predict unfavorable outcomes, such as sepsis, readmission and need of treatment escalation among patients with IAI..
Authors are kindly requested to emphasize the current concepts about these issues in the context of recent knowledge and the available literature. This articles should be quoted in the References list.
References
1. The critical role of pancreatic stone protein/regenerating protein in sepsis-related multiorgan failure. Front Med (Lausanne). 2023; 10: 1172529. Published 2023 May 5. doi:10.3389/fmed.2023.1172529.
2. Point-of-care pancreatic stone protein measurement in critically ill COVID-19 patients. BMC Anesthesiol. 2023;23(1):226. Published 2023 Jun 30. doi:10.1186/s12871-023-02187-w.
3. The Role of Pancreatic Stone Protein (PSP) as a Biomarker of Pregnancy-Related Diseases. J Clin Med. 2023; 12 (13): 4428. Published 2023 Jun 30. doi:10.3390/jcm12134428.
Minor editing of English language required
Author Response
Dear reviewer. thank you for your use

Reviewer 2 Report
A well designed and presented innovative research work on a novel marker which could improve diagnosis of sepsis. Please find below some comments/suggestions:
1)Coud you provide potential difference in PSP values between patients with and without bacteremia? As you mentioned at the discussion the role of PSP in bacteremia and cholangitis-the most frequent infection in your cohort.
2) Please give the reasons of re-admission: relapse of cholecystitis/cholangitis due to surgical issues? ( e.g need for operation which was not performed during the first hospitalization).
Author Response
dear reviewer thank you.
Find attached our response.

Reviewer 3 Report
1. The presented paper entitled "Prognostic Utility of Pancreatic Stone Protein in Patients with Intra-Abdominal Infection" presents a prospective observational cohort study conducted to evaluate the prognostic role of the Pancreatic Stone Protein (PSP) in patients admitted to the hospital with a diagnosis of intra-abdominal infection (IAI). The study included 40 adult patients consecutively admitted to the Internal Medicine Department of the University Hospital of Patras, Greece, with a diagnosis of IAI. PSP levels were measured within 24 hours since admission in whole blood. The results showed that PSP was an independent predictive factor for sepsis in the study group after adjusting for age, with an odds ratio of 7.888 (95% confidence interval: 1.247-49.890).
2. The findings of this study suggest that PSP can serve as a valuable prognostic tool in patients with IAI. The authors hyposese that it can predict unfavorable outcomes such as sepsis, readmission, and the need for treatment escalation. However, the limitation of a small sample size and the absence of a significant number of deaths in the cohort limit the ability to assess the prognostic value of PSP for mortality.
3. The number of clinical parameters (PSP, ferritin, CRP and fibrinogen) is not sufficient to answer the question posed by the authors of the article. Flow cytometric characterization of immune cell subpopulations is required.
4. Overall, the article represents an early-stage study for which a larger sample of patients is needed to narrow the ranges of the confidence intervals. Further research with a larger sample size and diverse patient population is warranted to validate these findings and explore the clinical implications of PSP in the management of intra-abdominal infections. As presented, the article is not particularly interesting due to the lack of statistical significance of the results.
Author Response
Dear reviewer thank you for your valuable comments

Round 2
Reviewer 3 Report
The authors made only minor cosmetic changes to the manuscript and did not significantly improve it. I was not satisfied with the authors' answers to my questions. The authors' efforts to improve the manuscript were not sufficient to recommend it for publication.
Author Response
Dear reviewer,
We made further changes to the manuscript. Changes are mentioned in red.
We made further English language editing
Thank you.